



# Flight Trajectory Optimization of Fly-Gen AWE Systems through a Harmonic Balance Method

Filippo Trevisi[1], Iván Castro-Fernández[2], Gregorio Pasquinelli[1], Carlo Emanuele Dionigi Riboldi[1], and Alessandro Croce[1]

[1]Department of Aerospace Science and Technology, Politecnico di Milano, Via La Masa 34, 20156 Milano, Italy
[2]Departamento de Bioingeniería e Ingeniería Aeroespacial, Universidad Carlos III de Madrid, Leganés, 28911 Madrid, Spain

**Correspondence:** F. Trevisi (filippo.trevisi@polimi.it)

**Abstract.** The optimal control problem for flight trajectories for Fly-Gen Airborne Wind Energy Systems (AWES) is a crucial research topic for the field, as suboptimal paths can lead to a drastic reduction in power production. One of the novelties of the present work is the expression of the optimal control problem in the frequency domain through a Harmonic Balance formulation. This allows to reduce the problem size by solving only for the main harmonics and to implicitly impose periodicity of the solution. The trajectory is described by the Fourier coefficients of the dynamics (elevation and azimuth angles) and of the control inputs (on-board wind turbines thrust and AWES roll angle). To isolate the effects of each physical phenomenon, optimal trajectories are presented with an increasing level of physical representation from the most idealized case: i) If the mean thrust power (mechanical power linked to the dynamics) is considered as the objective function, optimal trajectories are characterized by a constant AWES velocity over the loop and a circular shape. This is done by converting all the gravitational potential energy into electrical energy. At low wind speed, on-board wind turbines are then used as propellers in the ascendant part of the loop; ii) If the mean shaft power (mechanical power after momentum losses) is the objective function, a part of the potential energy is converted into kinetic and the rest into electrical energy. Therefore, the AWES velocity fluctuates over the loop; iii) If the mean electrical power is considered as the objective function, the on-board wind turbines are never used as propellers because of the power conversion efficiency. Optimal trajectories for case ii) and iii) have a circular shape squashed along the vertical direction. The optimal control inputs can be generally modelled with one harmonic for the on-board wind turbines thrust and two for AWES roll angle without a significant loss of power, demonstrating that the absence of high-frequency control is not detrimental to the power generated by Fly-Gen AWES.

## 1 Introduction

Airborne Wind Energy (AWE) is the branch of wind energy which aims at harvesting energy from the wind using airborne systems. Airborne Wind Energy Systems can be classified according to the flight operations, which are linked to the power generation technique. The flight operations can be divided into crosswind, tether-aligned and rotational, as discussed by Vermillion et al. (2021). Electrical power can be generated by a fixed or a moving ground station or, alternatively, it can be directly generated on-board and transmitted to the ground through the tether. The wing type, soft or fixed, additionally classifies the





AWES. This paper focuses of AWES based on a fixed-wing with on-board generation, known as Fly-Gen AWES. However, the methods developed can be applied to other AWE architectures, after an appropriate rework of the dynamic models. These methods are suitable for investigating the optimal trajectories of AWES and, especially when applied to low fidelity models, for understanding their physical characteristics. The interpretation of the physical characteristics of optimal trajectories and the analysis of how they are influenced by parameters describing the system and its operation is the main goal of this work. With this aim, solutions are compared with analytical solutions coming from first-principle models whenever possible.

The first analytical power equation of crosswind AWES was derived by Loyd (1980), and additional refinements, such as the one proposed by Trevisi et al. (2020a), made an effort to modify analytic equations to include gravitational and centrifugal effects. This kind of analytical models can be used to study how power and other relevant trends approximately scale with design parameters. However, they typically neglect the system dynamics and its effect on power generation. These effects can be studied with dynamical models, ranging from low to high fidelity. Typically, low- to mid-fidelity models are used to investigate optimal trajectories of AWE. Low-fidelity dynamic models are characterized by multiple assumptions, which simplify the models, and by the low computational cost. The quasi-steady model (van der Vlugt et al. (2019)) assumes the kite as a point mass in steady state for each point of the loop. This model is validated with experimental data (Schelbergen and Schmehl (2020)) and it is considered accurate for soft kites, where the inertia is low and the AWES quickly reaches the steady state. A similar approach is considered while deriving the Unicycle model (Fagiano et al. (2014); Vermillion et al. (2021)). Also this model, based on a point mass, is developed for soft-wing AWES and computes the velocity vector via quasi-steady flight equations. The kite orientation is found by a turning law that is derived from lateral force equilibrium and is validated through a number of experiments. The Unifoil model (Cobb et al. (2020)) is derived by modification of the Unicycle model in order to be applied to fixed-wing AWES. Indeed, the quasi-steady assumption is removed and the turning maneuvers modelled with a yaw dynamic equation.

Higher fidelity, but still computationally efficient, dynamic models are developed by Sánchez-Arriaga et al. (2017, 2019); Sánchez-Arriaga and Serrano-Iglesias (2021) as a part of the Lagrangian Kite Flight Simulators (LAKSA) package based on minimal coordinates, and by Gros and Diehl (2013) to study the dynamics of multiple AWES configurations. Moreover, thorough Newtonian dynamic models are used to compute reference flight paths and the consequent flight path control for soft-wing AWES (Fechner et al. (2015); Fechner and Schmehl (2016)) and for fixed-wing AWES (Licitra et al. (2019); Malz et al. (2019); Eijkelhof and Schmehl (2022)).

The dynamic models just introduced are particularly suitable to be used within optimal control studies for their computational inexpensiveness and for the reduced number of nonlinearities compared to even higher fidelity codes, such as kiteFAST (Jonkman et al. (2018)). The Unicycle and Unifoil models, introduced earlier, are mainly used to compute reference flight paths and for flight path control development (Cobb et al. (2020); Fernandes et al. (2021)). To ease the deployment of optimal control problems for AWE, *awebox* (awebox) is developed and used, for instance by Leuthold et al. (2018), Haas et al. (2019) and De Schutter et al. (2019), to solve optimal control problems. *awebox* solves optimal control problems in time, imposing periodicity constraints. A similar optimal control problem is studied by Horn et al. (2013), Malz et al. (2020a) and Malz et al. (2020b), where the optimal trajectory is found in time using a discretization by direct collocation and a homotopy strategy





based on the relaxation of the dynamic constraints (Gros et al. (2013)). Licitra et al. (2019) solved an optimal control problem
with an experimentally validated dynamic model of a Ground-Gen AWES. They find that, under some prescribed constraints,
circular and figure of eight trajectories produce similar mean power and that closed-loop control enhance robustness but decreases power production of about 10 %. Control in all operation phases is studied by Rapp et al. (2019) and Todeschini et al. (2021): the present work can be understood as a study of the guidance (or the reference trajectory) used during the power generation phase of their study.

Pasquinelli (2021) investigates the power losses in a circular trajectory with a dynamical quasi-analytical model. He finds that the causes of power losses are mainly two: the kite span non-perpendicularity with respect to the incoming wind during the motion and the AWES speed fluctuation over the loop. Makani team (Tucker (2020)) studies the flight trajectories of Fly-Gen AWES with a simplified quasi-analytical approach, aiming at describing their physical characteristics. They run their flight simulator for different trajectories and production strategies to derive analytical expressions, which can describe the consequences of different operational choices. Their production strategy at low wind speed is to convert part of the potential energy into kinetic and part into electrical, when the AWES moves downward. To reduce the potential energy exchange, they suggest to squash the trajectories along the vertical direction. Moreover, they explain that using electrical power to push the AWES upward is drastically decreasing the overall power production, as power needs to be converted from mechanical to electrical and again from electrical to mechanical, so that the related efficiencies are counted twice. They, in accord with the study for Ground-Gen by Stuyts et al. (2015), conclude that the electrical conversion losses should be considered when deciding on the production strategy. Following these conclusions, the present work also investigates the influence of the power generation efficiencies on the optimal trajectories.

As the aim of this work is to interpret optimal trajectories in a physical way, a low-fidelity dynamic model, similar to the one proposed by Fernandes et al. (2021) (reformulated for Fly-Gen AWES), is selected. Instead of solving the dynamics and the optimal control problem in time, the present approach models the problem in the frequency domain, making use of a Harmonic Balance method, which expands the periodic solution as a Fourier series (Lau et al. (1982); Pierre and Dowell (1985); Dimitriadis (2017)). Working with the Fourier coefficients and not with the time series themselves allows to reduce the problem size significantly, to look for periodic solutions implicitly and to study the solution in an intuitive way by looking at the contribution of the different harmonics. To the best of the authors' knowledge, this is the first work on AWES where an optimal control problem aided by a Harmonic Balance methodology is formulated.

Even though the frequency-domain formulation can be used for any periodic flight trajectories (i.e. circular and figure of eight), only circular trajectories are here analyzed to limit the paper scope and length. Figure of eight trajectories are intended to be analyzed and intensively compared with circular trajectories in a future work.

The paper is organized as follows: in Sect. 2 the flight dynamic model, the Harmonic Balance and the optimal control statement are introduced. In Sect. 3, the main results from steady state analytical models are recalled from literature, together with the introduction of some key non-dimensional numbers used later in the analyses. In Sect. 4, the solution obtained with the Harmonic Balance formulation is validated against the time integration. In Sect. 5, optimal control problems with constant wind inflow and no constraints on the mean elevation angle is analyzed. This extreme idealization allows for the understanding



of some optimal trajectory characteristics which are also present in more realistic cases. Section 6 focuses on the results of a

more realistic optimal control problem. Indeed, the wind shear and a constraint on the minimum elevation angle are included

in the analyses. Finally, in Sect. 7 the results are discussed and the main conclusions summarized.

## 2  Methodology

### 2.1  Flight Dynamic Model

Two coordinate systems (Figure 1a) are defined to derive the equations of motion. The ground coordinate system (denoted by

$\mathcal{F}_G$) is inertial and centered at the ground station: $\mathbf{e}_x$ points downwind, $\mathbf{e}_z$ toward the Zenith and $\mathbf{e}_y$ completes the right-handed

frame. For convenience, spherical coordinates are used to describe the position of the airborne unit with $L_t$ the tether length, $\phi$

the azimuth angle and $\beta$ the elevation angle. The spherical reference frame (denoted by $\mathcal{F}_S$) is unequivocally defined at every

position with the origin at the AWES center of mass, $\mathbf{e}_r$ pointing outward the sphere in the radial direction, $\mathbf{e}_\phi$ normal to $\mathbf{e}_r$

and contained on a plane parallel to $x - y$ and $\mathbf{e}_\beta = \mathbf{e}_r \times \mathbf{e}_\phi$. The position $\mathbf{p}$, velocity $\mathbf{v}$ and acceleration $\mathbf{a}$ projected into the

spherical reference frame $\mathcal{F}_S$ are

$$
\begin{aligned}
\mathbf{p} &= L_t \mathbf{e}_r, \\
\mathbf{v} &= L_t \dot{\phi} \cos\beta \, \mathbf{e}_\phi + L_t \dot{\beta} \mathbf{e}_\beta, \\
\mathbf{a} &= \left( -L_t \dot{\phi}^2 \cos^2\beta - L_t \dot{\beta}^2 \right) \mathbf{e}_r + (L_t \ddot{\phi} \cos\beta - 2L_t \dot{\phi}\dot{\beta}\sin\beta)\mathbf{e}_\phi + \left( L_t \dot{\beta}^2 \sin\beta\cos\beta + L_t \ddot{\beta} \right) \mathbf{e}_\beta.
\end{aligned}
\tag{1}
$$

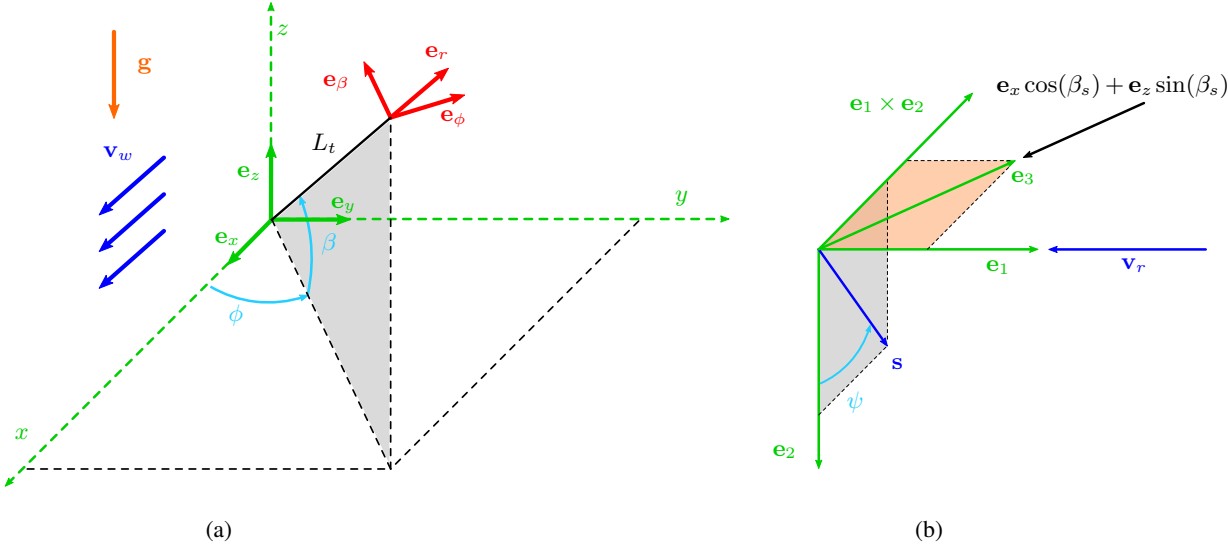

(a)                                                                  (b)

**Figure 1. (a)** Ground reference frame $\mathcal{F}_G$ ($\mathbf{e}_x - \mathbf{e}_y - \mathbf{e}_z$) and spherical reference frame $\mathcal{F}_S$ ($\mathbf{e}_r - \mathbf{e}_\phi - \mathbf{e}_\beta$) and **(b)** sketch for the spanwise unit vector $\mathbf{s}$ definition.



The wind velocity is in the positive $x$-axis direction of $\mathcal{F}_G$ and projected into the spherical reference frame is

$$\mathbf{v}_w = v_w \left( \cos\phi\cos\beta\mathbf{e}_r - \sin\phi\mathbf{e}_\phi - \cos\phi\sin\beta\mathbf{e}_\beta \right), \qquad v_w(h) = v_{w,0} \left( \frac{h}{h_0} \right)^{\alpha_s} = v_{w,0} \left( \frac{L_t}{h_0}\sin\beta \right)^{\alpha_s}, \tag{2}$$

where the wind speed $v_w$ as function of the altitude $h$ is modelled with an exponential law: $v_{w,0}$ is the reference wind speed at
the reference altitude $h_0$ and $\alpha_s$ is the wind shear exponent. The relative speed between the AWES and the wind is

$$\mathbf{v}_r = \mathbf{v} - \mathbf{v}_w. \tag{3}$$

To describe the AWES attitude, a non slideslip velocity constraint is included in the modelling. Indeed, the wing operates at the highest performance under this condition. To impose this constraint implicitly, the unit vector $\mathbf{e}_1$ is defined to point the opposite direction of the relative wind speed

$$\mathbf{e}_1 = -\frac{\mathbf{v}_r}{|\mathbf{v}_r|}. \tag{4}$$

The spanwise unit vector $\mathbf{s}$ (with origin at the center of mass and pointing in the right-wing span direction) is defined perpendicular to $\mathbf{e}_1$ with the procedure illustrated in Fig. 1b. A second vector $\mathbf{e}_3$ is defined as a unit vector a plane parallel to $x - z$ plane with elevation $\beta_s$ (and negative sign)

$$\mathbf{e}_3 = -(\mathbf{e}_x\cos(\beta_s) + \mathbf{e}_z\sin(\beta_s)), \quad \mathbf{e}_x = \cos\phi\cos\beta\mathbf{e}_r - \sin\phi\mathbf{e}_\phi - \cos\phi\sin\beta\mathbf{e}_\beta, \quad \mathbf{e}_z = \sin\beta\mathbf{e}_r + \cos\beta\mathbf{e}_\beta. \tag{5}$$

Note that $\mathbf{e}_3$ points upwind when $\beta_s = 0$. The unit vector $\mathbf{e}_2$ is then defined as

$$\mathbf{e}_2 = \frac{\mathbf{e}_3 \times \mathbf{e}_1}{|\mathbf{e}_3 \times \mathbf{e}_1|}, \tag{6}$$

where $|\mathbf{e}_3 \times \mathbf{e}_1|$ can take values smaller than one because $\mathbf{e}_3$ and $\mathbf{e}_1$ are not defined to be perpendicular in general. In this way, $\mathbf{e}_2$ is perpendicular to the plane $\mathbf{e}_3$-$\mathbf{e}_1$. Rodrigues' formula is then used to define $\mathbf{s}$ through a rotation of $\psi$ around $\mathbf{e}_1$, starting from $\mathbf{e}_2$

$$\mathbf{s} = \mathbf{e}_2\cos\psi + (\mathbf{e}_1 \times \mathbf{e}_2)\sin\psi + \mathbf{e}_1(\mathbf{e}_1 \cdot \mathbf{e}_2)(1 - \cos\psi). \tag{7}$$

With this formulation, $\mathbf{s}$ is defined to be always perpendicular to the relative wind and its components are defined by a unique angle $\psi$, called hereafter roll angle. When $\psi = 0$, $\mathbf{s}$ is perpendicular to $\mathbf{e}_3$.

The aerodynamic lift $\mathbf{L}$ and the drag $\mathbf{D}$ take the standard form

$$\mathbf{L} = \frac{1}{2}\rho A C_L |\mathbf{v}_r| \mathbf{v}_r \times \mathbf{s}, \qquad \mathbf{D} = \frac{1}{2}\rho A C_D |\mathbf{v}_r| \mathbf{v}_r, \tag{8}$$

where $\rho$ is the air density, $A$ is the wing area and the lift and drag coefficients $C_L$ and $C_D$ are considered constant. The drag coefficient $C_D$ includes the contribution from the tether drag (Trevisi et al. (2020a)). The gravitational force $\mathbf{F}_g$ and the tether force $\mathbf{T}$ are

$$\mathbf{F}_g = -mg(\sin\beta\mathbf{e}_r + \cos\beta\mathbf{e}_\beta), \qquad \mathbf{T} = -T\mathbf{e}_r, \tag{9}$$





where $m$ is the AWES mass, $g$ the gravitational acceleration and $T$ the absolute value of the tether force. The thrust produced
by the on-board wind turbines $\mathbf{D}_t$ is expressed as a linear function of the aerodynamic drag with gain $\gamma$

$$\mathbf{D}_t = \gamma \mathbf{D}. \tag{10}$$

The dynamic equations of motion in compact form read

$$m\mathbf{a} = \mathbf{L} + \mathbf{D} + \mathbf{D}_t + \mathbf{F_g} + \mathbf{T}, \tag{11}$$

recalling that $\mathbf{a}$ is given by Eq. 1.

As the objectives of the optimal control problems are linked to the power production, three different power quantities are
defined. The thrust power $P_t$ (i.e. the power linked to the AWES dynamics) is estimated as a dot product of $\mathbf{D}_t$ and the relative
velocity

$$P_t = \mathbf{D}_t \cdot \mathbf{v}_r. \tag{12}$$

The shaft power $P_s$ (i.e. the mechanical power that can be converted to electrical power) is modelled using 1D momentum
theory (actuator disc) as

$$P_s = (1-a)P_t = \left( \frac{1}{2} + \frac{1}{2}\sqrt{1 - \gamma C_D \frac{A}{A_t}} \right) P_t, \tag{13}$$

where the induction $a$ is found by setting the thrust given by momentum theory[1] equal to $\mathbf{D}_t$, as in Trevisi et al. (2020b), and
$A_t$ is the total turbine area.

Finally, the electrical power exchanged with the grid $P$ takes into account the generator and transmission efficiency $\eta_{el}$

$$P = \begin{cases} P_s - (1-\eta_{el})P_s & \text{for } \gamma \geq 0 \\ P_s + (1-\eta_{el})P_s & \text{for } \gamma < 0. \end{cases} \tag{14}$$

When power is generated ($\gamma > 0$), the electrical power distributed to the grid $P$ is lower than the shaft power $P_s$ because
of electrical efficiencies. When power from the grid is used, the electrical power requested to the grid $P$ is instead higher in
absolute value compared to the shaft power $P_s$. To model the discontinuity in a continuous optimization framework, the logistic
function is used

$$P = P_s - \left( \frac{1 - e^{-f\gamma}}{1 + e^{-f\gamma}} \right)(1 - \eta_{el})P_s, \tag{15}$$

where $f$ is taken equal to 100.

---

[1]$\mathbf{T}_d = \frac{1}{2}\rho A_t (4a(1-a))|\mathbf{v}_r|\mathbf{v}_r$



## 2.2 Frequency Domain Formulation

Frequency domain formulations present clear advantages when solving for periodic solutions of dynamic and control problems. They have the capability of solving for both stable and unstable (unlike time integration methods) branches of periodic solutions in an efficient way. Moreover, they use less variables to describe the same problems. Since the problem of optimal trajectories for AWES has a periodic nature, the flight dynamic model just introduced is expressed in the frequency domain. The Harmonic Balance methodology is then used to transform the differential equations of motion into a set of nonlinear algebraic equations (Dimitriadis (2017)). The equations of motion (Eq. 11) can be written as a set of second-order nonlinear differential equations in the form

$$\boldsymbol{f}(\mathbf{x}, \dot{\mathbf{x}}, \ddot{\mathbf{x}}, \mathbf{u}) = \mathbf{0}, \qquad \mathbf{x} = \left[\beta(t),\ \phi(t)\right]^T, \qquad \mathbf{u} = \left[\psi(t),\ \gamma(t)\right]^T, \tag{16}$$

where $\mathbf{x}$ is the state vector and $\mathbf{u}$ is the control vector. By assuming that Eq. 16 accepts periodic solutions, every variable of the state vector is expanded as a Fourier series of order $N_x$

$$x(t) \approx \frac{X_0}{2} + \sum_{k=1}^{N_x} X_{k,s}\sin(k\omega t) + X_{k,c}\cos(k\omega t), \quad \mathbf{X} = \left[X_0,\ X_{1,s},\ X_{2,s},\ ...X_{1,c},\ X_{2,c},\ ...\right]^T, \tag{17}$$

with $\omega = \dfrac{2\pi}{\mathcal{T}}$ being the fundamental frequency of the motion and $\mathcal{T}$ the period. Alternatively, the state vector can be expressed as

$$x(t) \approx \frac{A_0}{2} + \sum_{k=1}^{N_x} A_k\cos(k\omega t - \theta_k), \quad \mathbf{A} = \left[A_0,\ A_1,\ ...\right]^T, \quad \theta = \left[\theta_1,\ \theta_2,\ ...\right]^T, \tag{18}$$

where

$$A_0 = X_0, \qquad A_k = \sqrt{X_{k,s}^2 + X_{k,c}^2}, \quad \theta_k = \arctan\left(\frac{X_{k,s}}{X_{k,c}}\right). \tag{19}$$

The first and second time derivatives of the state vector can be found analytically

$$\dot{x}(t) \approx \sum_{k=1}^{N_x} k\omega\left(X_{k,s}\cos(k\omega t) - X_{k,c}\sin(k\omega t)\right), \quad \ddot{x}(t) \approx -\sum_{k=1}^{N_x}(k\omega)^2\left(X_{k,s}\sin(k\omega t) + X_{k,c}\cos(k\omega t)\right). \tag{20}$$

Similarly, the control inputs, assumed to be periodic, can also be expressed as a Fourier series of order $N_u$

$$u(t) \approx \frac{U_0}{2} + \sum_{k=1}^{N_u} U_{k,s}\sin(k\omega t) + U_{k,c}\cos(k\omega t), \quad \mathbf{U} = \left[U_0,\ U_{1,s},\ U_{2,s},\ ...U_{1,c},\ U_{2,c},\ ...\right]^T, \tag{21}$$

where $N_u \le N_x$. By introducing Eqs. 17, 20 and 21 into Eq. 16, the equations of motion can be expanded as a Fourier series of order $N_x$

$$\mathbf{f}(\mathbf{X}_\beta, \mathbf{X}_\phi, \mathbf{U}_\psi, \mathbf{U}_\gamma, \omega, t) \approx \frac{\mathbf{F}_0}{2} + \sum_{k=1}^{N_x} \mathbf{F}_{k,s}\sin(k\omega t) + \mathbf{F}_{k,c}\cos(k\omega t) = \mathbf{0}, \quad \mathbf{F} = \left[\mathbf{F}_0,\ \mathbf{F}_{1,s},\ \mathbf{F}_{2,s},\ ...\mathbf{F}_{1,c},\ \mathbf{F}_{2,c},\ ...\right]^T. \tag{22}$$





The Fourier coefficients of the equations of motion are found numerically by applying the Fourier coefficient definition to the time series, which should have a minimum size of $2N_x + 1$. Note that a continuous switch between time and frequency domain is needed. A Galerking methodology is then applied by pre-multiplying Eq. 22 by 1, $\sin(k\omega t)$, $\cos(k\omega t)$ and subsequently integrating the resulting equation over one period. The result is a set of $2 \times (2N_x + 1)$ nonlinear algebraic equations as a consequence of the orthogonality properties of the selected basis of trigonometric functions

$$\mathbf{R}(\mathbf{X}_\beta, \mathbf{X}_\phi, \mathbf{U}_\psi, \mathbf{U}_\gamma, \omega) = \mathbf{0}, \tag{23}$$

which can be understood as the residuals of the equations of motion expressed in the frequency domain. For given periodic control inputs and a given fundamental frequency, the periodic solution can be found by looking for the Fourier coefficients $(\mathbf{X}_\beta, \mathbf{X}_\phi)$ of the dynamics which solve Eq. 23.

## 2.3 Optimal Control Problem (OCP)

In this work, the frequency-domain formulation is included within an optimal control problem (OCP). A generic optimization problem can be written as

$$
\begin{aligned}
\mathcal{X}^* \quad &= \min_{\mathcal{X}} obj(\mathcal{X}), \\
\text{s.t.:} \quad &\mathbf{lb} \leq \mathcal{X} \leq \mathbf{ub} \\
&\mathbf{g}(\mathcal{X}) \leq 0 \\
&\mathbf{h}(\mathcal{X}) = 0,
\end{aligned}
\tag{24}
$$

where $\mathcal{X}$ are the unknown optimization variables, $\mathcal{X}^*$ their optimal values, $obj$ the objective function, $\mathbf{lb}$ and $\mathbf{ub}$ the lower and upper bounds of $\mathcal{X}$, $\mathbf{g}$ the inequality and $\mathbf{h}$ the equality constraints. In the present formulation, the optimization variables are the Fourier coefficients of the state variables, of the control inputs and the fundamental frequency

$$\mathcal{X} = [\mathbf{X}_\beta; \mathbf{X}_\phi; \mathbf{U}_\psi; \mathbf{U}_\gamma; \omega]. \tag{25}$$

The negative value of the mean thrust power $\hat{P}_t$ (Eq. 12), shaft power $\hat{P}_s$ (Eq. 13) or electric power $\hat{P}$ (Eq. 15) over the loop is taken as objective function, where the symbol $\hat{\cdot}$ stands for the mean value over the loop. The equality constraints are the aggregation of the residuals of the equation of motion in the frequency domain $\mathbf{R}$ (Eq. 23) and additional physical constraints $\mathbf{r}$ (e.g. certain quantities can be imposed to be constant over the loop)

$$\mathbf{h}(\mathcal{X}) = [\mathbf{R}(\mathcal{X}); \mathbf{r}(\mathcal{X})] = \mathbf{0}. \tag{26}$$

Inequality constraints $\mathbf{g}$ can also be included in the problem (e.g. the minimum elevation angle over the loop can be bounded). A graphical representation of the OCP setup is given in Figure 2. One of the advantages of the frequency formulation is that the derivatives of flight dynamic model with respect to the optimization variables can be taken analytically and provided to the solver, allowing for a deep and fast convergence of the solution. The OCP is implemented in *MATLAB*® environment and solved with the interior-point algorithm implemented in *fmincon*. As the chosen optimization algorithm (gradient-based) can



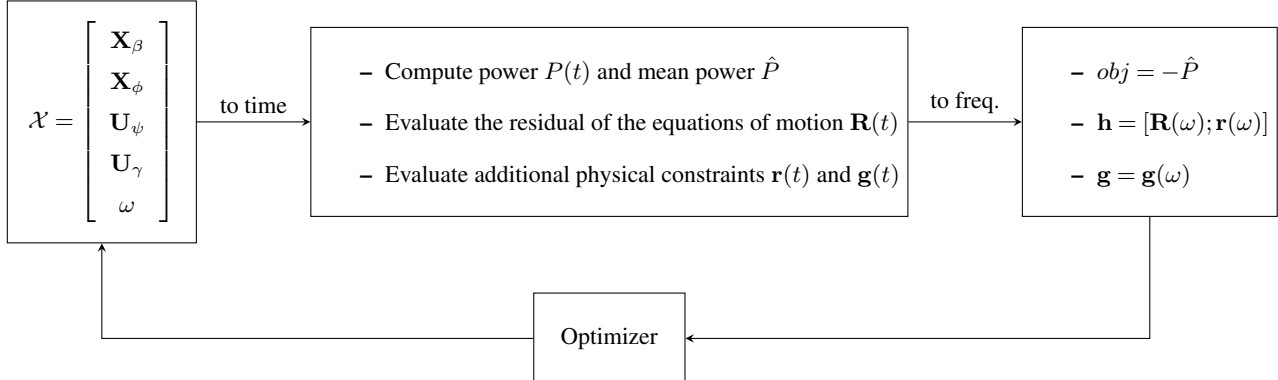

**Figure 2.** Graphical representation of the optimal control problem setup.

only look for local optima, the initial guess may influence the solution. In this work, the initial guesses are taken to be circular trajectories, leading to circular shaped optimal trajectories. Figure of eight trajectories can be implemented as initial guesses,

which may lead to figure of eight shaped optimal trajectories. A detailed comparison between these two trajectory types is left for future works.

## 3 Steady State Model

To compare the results of the optimal control problem with idealized analytical expressions, the main results from a refined version of the Loyd power equation (Loyd (1980)) are here briefly recalled. The thrust power equation, with the assumption of

linear crosswind motion (Trevisi et al. (2020b)) is

$$P_{t,L} = \frac{1}{2}\rho\,A\,V_w^3\,\gamma\,\frac{C_L}{G}\left(1+\left(\frac{G}{1+\gamma}\right)^2\right)^{3/2} = \frac{1}{2}\rho\,A\,V_w^3\,\gamma\,\frac{C_L}{G}\left(1+G_t^2\right)^{3/2}, \tag{27}$$

where the system glide ratio (including tether drag) is $G = \frac{C_L}{C_D}$ and, for readability, a modified glide ratio is defined as $G_t = \frac{C_L}{C_D(1+\gamma)}$ by including the drag of the on-board propellers. The shaft power takes into account the on-board wind turbine induction $a$

$$P_{s,L} = (1-a)P_{t,L} = \left(\frac{1}{2}+\frac{1}{2}\sqrt{1-\gamma C_D\frac{A}{A_t}}\right)P_{t,L}. \tag{28}$$

Finally, the power generated and sent to the grid takes into account the efficiencies of the electrical conversion,

$$P_L = \eta_{el}P_{s,L} = \eta_{el}\left(\frac{1}{2}+\frac{1}{2}\sqrt{1-\gamma C_D\frac{A}{A_t}}\right)\frac{1}{2}\rho\,A\,V_w^3\,\gamma\,\frac{C_L}{G}\left(1+G_t^2\right)^{3/2}. \tag{29}$$

For high $G$, the power equation simplifies to

$$P_L \approx \frac{1}{2}\rho\,A\,V_w^3\,C_L\,G^2\,\eta_{el}\left(\frac{1}{2}+\frac{1}{2}\sqrt{1-\gamma C_D\frac{A}{A_t}}\right)\frac{\gamma}{(1+\gamma)^3}. \tag{30}$$





For this expression, the value of $\gamma$ which maximizes the power is only a function of the non-dimensional quantity $C_D \frac{A}{A_t}$. In Figure 3a, the electrical power $P_L$, normalized with the electrical power at $C_D \frac{A}{A_t} = 0$, is plotted as a function of $\gamma$ and $C_D \frac{A}{A_t}$. For increasing values of $C_D \frac{A}{A_t}$, the values of $\gamma$ which maximizes power production decreases. The maximum normalized power as a function of $C_D \frac{A}{A_t}$ is shown in Fig. 3b, highlighting that the analytical expression predicts a decrease in power production for increasing $C_D \frac{A}{A_t}$.

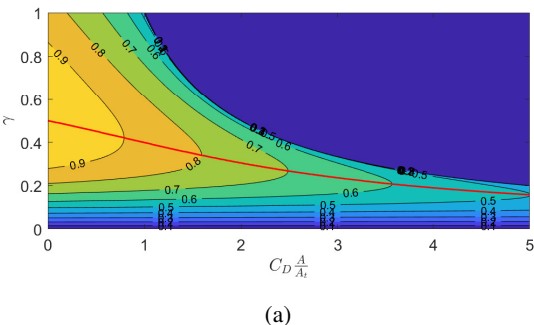
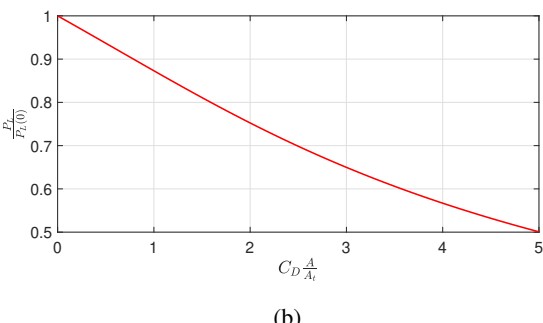

(a)                                                                        (b)

**Figure 3. (a)** Normalized power $\frac{P_L}{P_L(C_D \frac{A}{A_t} = 0)}$ as a function of $C_D \frac{A}{A_t}$ and $\gamma$ and **(b)** its maximum value as a function of $C_D \frac{A}{A_t}$ for high glide ratios $G$.

The tether force can be evaluated as

$$T_L = \frac{1}{2} \rho \, A \, V_w^2 \, \frac{C_L}{G_t} \left(1 + G_t^2\right)^{3/2}. \tag{31}$$

Trevisi et al. (2020a) showed that for high $G$, neglecting gravity and with constant incoming wind, it exists an opening angle $\tilde{\Phi}$ (angle swept by the AWES during the circular trajectory, see Fig. 4) which erases the power losses due to centrifugal forces and that it is only a function of the non-dimensional mass parameter

$$M = \frac{m}{\frac{1}{2}\rho A C_L L_t}. \tag{32}$$

In this idealized case, the turning radius is $R = L_t \sin \tilde{\Phi}$ and the revolution period is

$$\mathcal{T}_L = \frac{2\pi R}{v_w G_t} \tag{33}$$

In addition to the non-dimensional mass parameter, the Froude number, which weights the fluid inertial forces to gravity forces, is used in this work

$$F_r = \sqrt{\frac{v_w^2}{g \cdot L_t}}, \tag{34}$$

where the reference velocity is the wind velocity and the reference length is the tether length. By combining the previously introduced non-dimensional parameters, the gravity ratio $G_r$ is defined as

$$G_r = \frac{M}{F_r^2 G_t^2} = \frac{mg}{\frac{1}{2}\rho A C_L v_w^2 G_t^2}, \tag{35}$$





which represents the ratio between gravitational force and aerodynamic lift, similarly to the one introduced in Pasquinelli
(2021).

In the following sections, the results will be generalized as a function of the non-dimensional parameters just introduced.
Input parameters from Makani *MX2* design (Tucker (2020)) will be used as reference values to present the results (Tab. 1).

**Table 1.** Reference values for the examples (Values from the Makani *MX2* description Tucker (2020)), associated non-dimensional parameters
and quantities evaluated with the steady state model for $\gamma$ maximizing Eq. 29.

| $m$ | 2000 kg | $A$ | 54 m$^2$ | $A_t$ | 35 m$^2$ | $L_t$ | 300 m | $C_L$ | 1.8 | $C_D$ | 0.15 |
|---|---|---|---|---|---|---|---|---|---|---|---|
| $\eta_{el}$ | 0.8 | $\rho$ | 1.225 kg m$^{-3}$ | $g$ | 9.81 m s$^{-2}$ | $v_w$ | 6 m s$^{-1}$ | | | | |
| $G$ | 12 | $M$ | 0.1120 | $C_D\frac{A}{A_t}$ | 0.231 | $\gamma$ | 0.488 | $a$ | 0.029 | $G_t$ | 8.06 |
| $F_r$ | 0.1106 | $G_r$ | 0.1408 | $P_L$ | 218.0 kW | $T_L$ | 142.6 kN | $|\mathbf{v}_k|_L$ | 48.4 m s$^{-1}$ | $\mathcal{T}_L$ | 12.6 s |

## 4 Validation of the Frequency-Domain Formulation against Time Integration

To make sure the frequency-domain formulation is well implemented and finds solutions which respect the equations of motion,
they are compared with the solution coming from a time integration scheme. The model described in Sect. 2.1 is solved with
the *MATLAB*® *ode45* integration scheme. After solving the periodic solution with the Harmonic Balance methodology, the
Fourier coefficients of the state and control vector are retrieved. The state vector at $t = 0$ is used as an initial condition for
the numerical integration. The control inputs must be computed from their Fourier series at every step of the integration. In
Appendix A, a comparison for a circular and a figure of eight trajectory is shown. The solution of the dynamics, represented
by the azimuth and elevation, for the two cases is equivalent demonstrating that the frequency-domain formulation is accurate
enough to be used in the present optimal control problem framework.

## 5 Optimal Control Problems with Constant Inflow and no Elevation Angle Constraints

As the analysis is limited to circular trajectories, a cylindrical reference frame $\mathcal{F}_C$, similar to the one employed in Trevisi et al.
(2020a), Trevisi et al. (2021) and Pasquinelli (2021), is used to present the results. A graphical representation of $\mathcal{F}_C$ is given in
Figure 4. The longitudinal axis of $\mathcal{F}_C$ is aligned with the mean elevation angle $\hat{\beta}$. The angle $\beta_m$ denotes the minimum elevation
angle and $\Phi$ the opening angle. The angular position of the AWES is defined by $\alpha$ and when $\alpha = 0$ the kite moves upward (i.e.
$\dot{\alpha} > 0$).

To increase complexity incrementally, the optimal control problems (OCPs) are modified from the most idealized case to a
realistic one. For the idealized cases analyzed in this section, uniform incoming wind speed ($\alpha_s = 0$) and no minimum elevation
angle constraints are considered. In this section, $\beta_s$ (Eq. 5) is set equal to zero, such that $\mathbf{e}_3$ points upwind. In this way, when
the roll is equal to zero ($\psi = 0°$), the span direction is perpendicular to the incoming wind.





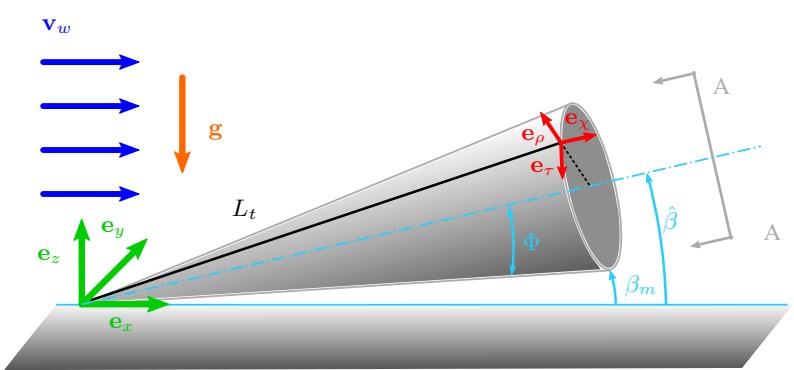

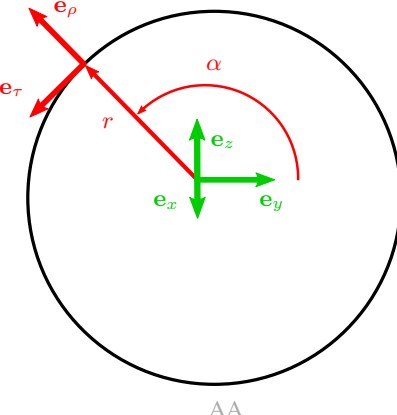

**Figure 4.** Cylindrical reference system $\mathcal{F}_C$ used to analyze circular trajectories.

## 5.1 Optimizing for the Mean Electrical Power in absence of Gravity

For the most idealized case, the gravity is null $g = 0$, such that $F_r \rightarrow \infty$ and $G_r = 0$. The objective function is taken as the mean electrical power, given in Eq. 15. By solving the OCP for the example (Table 1), it is found that the solution has constant

270 values over the trajectory and the average power output is equal to the one evaluated with the analytical expression in Eq. 29. Figure 5a shows the evolution of $\beta$ and $\phi$, highlighting that the solution is a circle. Due to the constant values of the solution, quantities such as tether force along the axial symmetry axis, $\gamma$, AWES velocity and others can be found with the formulation assuming a crosswind straight motion, as Sect. 3.

For the solution to be optimal, it is found that the AWES span is perpendicular to the wind speed, or, in analytical terms, that

275 $\psi = 0$. Figure 5b shows the optimal opening angle $\Phi^*$ as a function of a modified non-dimensional mass parameter $M_t$ found by solving a number of OCPs with different $G$ ($G \in [10\ 30]$), $M$ ($M \in [0.025\ 0.15]$) and $C_D \frac{A}{A_t}$ ($C_D \frac{A}{A_t} \in [0\ 0.4]$).

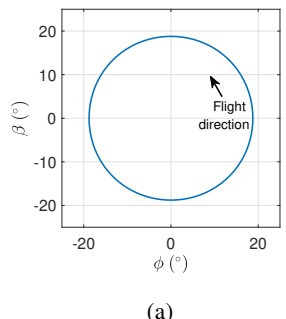

(a)

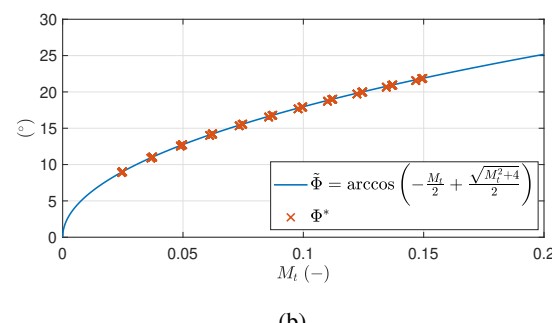

(b)

**Figure 5. (a)** Optimal trajectory and **(b)** optimal opening angles $\Phi$ (x) found by solving multiple OCPs and analytical expression (-) as a function of the modified non-dimensional mass parameter $M_t$.



The values of $\Phi^*$ can be accurately described by

$$\tilde{\Phi} = \arccos\left(-\frac{M_t}{2} + \frac{\sqrt{M_t^2 + 4}}{2}\right), \qquad M_t = \frac{M}{1 + \frac{1}{G_t^2}}, \qquad (36)$$

which for high glide ratios coincides with the analytical formulation given in Trevisi et al. (2020a).

### 5.2 Optimizing for the Mean Thrust Power considering Gravity

Gravity is now included in the modelling and the objective function is taken as the mean thrust power $\hat{P}_t$ (Eq. 12). The results of two slightly different OCPs are shown for the sake of understanding the results and they are summarized in Table 2. In OCP A, the control inputs are modelled with 5 harmonics. In OCP B, the time series of the control input $\psi$ is modelled as a constant and only one harmonic is used for the control input $\gamma$. Additionally, the absolute value of the AWES velocity is imposed to be constant (additional equality constraint). As the control inputs act up to the first harmonic, this constraint is set by imposing the first Fourier coefficients of the AWES velocity to zero, while no constraints are imposed on the higher-order harmonics. In Table 2, the mean thrust power (objective function) is also reported and compared with the analytical formulation (Eq. 27). The objective function of the two OCPs is almost the same, showing that the two problems are basically equivalent.

**Table 2.** Settings of the two optimal control problems maximizing the mean thrust power considering gravity.

| OCP | $N_x$ | $N_\gamma$ | $N_\psi$ | size **ov** | additional constr. | size **h** | $\hat{P}_t$ (kW) | $\mathcal{T}$ (s) |
|---|---|---|---|---|---|---|---|---|
| A | 10 | 5 | 5 | 65 | - | 42 | 285.4 | 12.8 |
| B | 10 | 1 | 0 | 47 | $|\mathbf{v}_k| = const$ | 44 | 284.5 | 12.7 |
| L | | | | analytical model | | | 280.4 | 12.8 |

By solving the OCPs, it is found that the optimal solutions have a negative mean elevation of $\hat{\beta}_A \approx -8.2\,°$ and $\hat{\beta}_B \approx -7.8\,°$. The trajectories, shown in Figure 6a, have a circular shape (a circle with radius $\tilde{\Phi}$ is marked as $-.$), but it is not any more a perfect circle. Figure 6b shows the trends of the control input $\psi$ as a function of $\alpha$ (see Figure 4 for definition). For OCP A, it fluctuates with small amplitude about the mean value, which is close to zero. For this reason, it is modelled as a constant in OCP B. The optimal constant value is also close to zero, meaning that the AWES span is perpendicular to the wind speed direction. Since the two OCPs present similar optimal values of power, it is found that the optimal solutions are not sensitive to the fluctuations of the roll angle $\psi$.

Figure 7a shows the evolution of $\gamma$ as a function of $\alpha$. The mean values for the two OCPs are close to the value maximizing Eq. 27, denoted in figure as $\gamma_L$. $\gamma$ takes values higher than the mean in the descending leg of the loop and negative values in the ascendant leg. This means that in the ascendant leg the on-board wind turbines are operated as propellers. Figure 7b shows the absolute value of the optimal AWES velocity $|\mathbf{v}_k|$ for the two OCPs. In OCP B, this value is constrained to be a constant by imposing null only its first harmonic. Since it is not possible to impose constraints at frequencies where the control ($\gamma$ and $\psi$) is not acting, higher-order harmonics are not constrained. The mean values of $|\mathbf{v}_k|$ are similar the one predicted by the



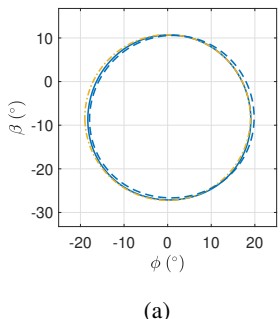

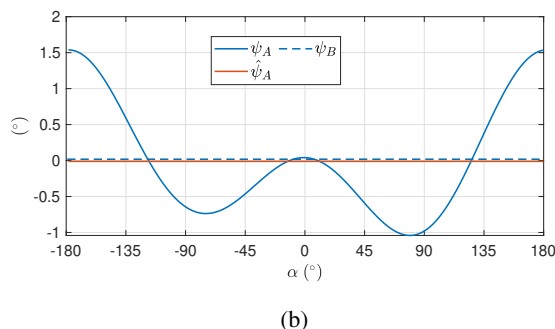

(a)

(b)

**Figure 6. (a)** Optimal trajectory for OCP A (-), B (- -) and a circle with radius $\tilde{\Phi}$ (−.) and **(b)** optimal $\psi$ as a function of the angular position.

steady model. Since the fluctuations for OCP A are small compared to the magnitude of the absolute value, the influence of the velocity fluctuations on the overall performances is investigated in OCP B, showing that they impact weakly the optimal solution.

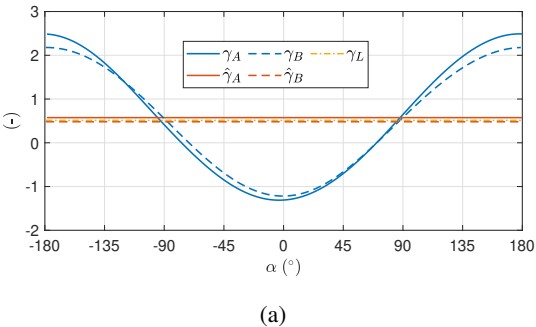

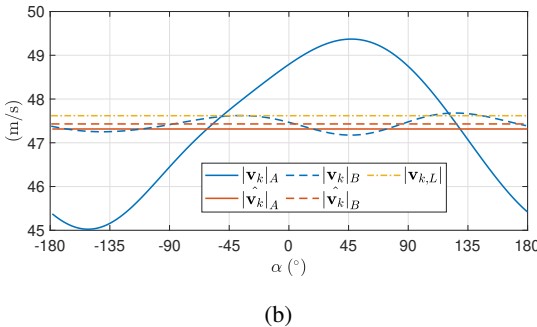

(a)

(b)

**Figure 7. (a)** Optimal $\gamma$ and **(b)** absolute value of the optimal AWES velocity $|\mathbf{v}_k|$ as a function of the angular position.

305    Figure 8a shows the magnitude of the tether force for the two cases. As it scales with the relative wind speed squared, also the tether force has an almost constant trend (the fluctuations are small compared to the mean). To compare the two OCPs and draw some conclusions, the power output, shown in Figure 8b, is to be analyzed. The mean thrust power output for the two OCPs is slightly higher than $P_{t,L}$. This is due to a non linear effect induced by the combination of gravity and mean elevation angle different from zero as compared to the idealized case. Indeed, for negative mean elevation, the combination leads to an

310    increase of mean thrust power, while the opposite occurs in case of positive mean elevation. As the effects on power is almost negligible and it does not primarily impact the main outcomes of this paper, a detailed explanation of this phenomenon is here avoided, but the reader can find more details in Pasquinelli (2021). The theoretical thrust power, given in Eq. 27, is derived neglecting gravity. However, it approximates well the power output obtained through the OPC, which includes gravity.

        As the two analyzed OCPs are almost equivalent, the optimal trajectories are characterized by the perpendicularity of the

315    AWES span with respect to the wind ($\psi = 0$) and a constant AWES velocity. In order to keep the AWES velocity constant over



the loop, the on-board wind turbines balance the action of the gravitational force. In the descendent leg, the on-board wind turbines harvest the gravitational potential energy and in the ascendant leg, that power is given back to the system.

Following these considerations, the power trend, as shown in Figure 8b, can then be approximated as

$$P_t(\alpha) \approx P_{t,L} + m\mathbf{g} \cdot \mathbf{v}_k \approx P_{t,L} - mgv_wG_t\cos\alpha. \tag{37}$$

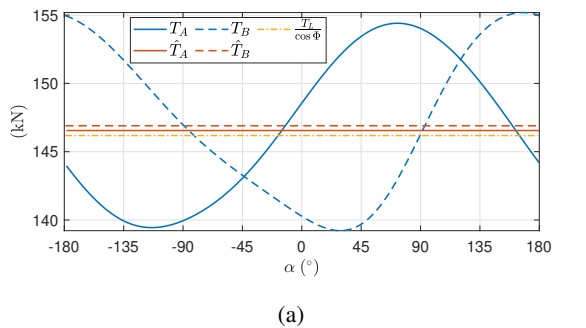
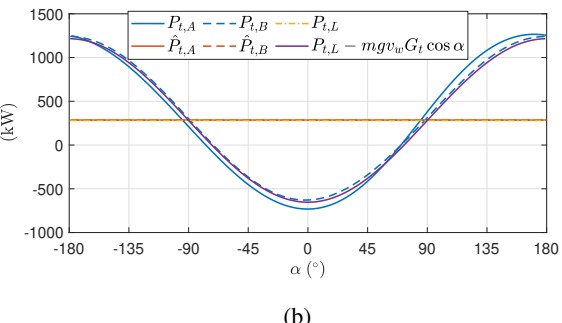

(a)                                         (b)

**Figure 8. (a)** Tether force $T$ and **(b)** optimal thrust power production and consumption $P_t$ as a function of the angular position.

The on-board wind turbines thrust can be approximated with $\mathbf{D}_t \approx (\hat{\gamma} + A_{\gamma,1}\cos(\alpha - \theta_{\gamma,1}))\mathbf{D}$, where $\mathbf{D}$ is constant because the AWES velocity is found to be constant and from Fig. 7a it is found that $\theta_{\gamma,1} \approx 180°$. As the thrust power can be written as the product of $\mathbf{D}_t$ and the relative wind speed $\mathbf{v}_r$ (Eq. 12), the amplitude of the first Fourier coefficient of $\gamma$, considering Eq. 37, can be approximated by

$$A_{\gamma,1} \approx \frac{m\mathbf{g} \cdot \mathbf{v}_k}{-\cos\alpha\,\mathbf{D} \cdot \mathbf{v}_r} \approx \frac{mg}{\frac{1}{2}\rho AC_D v_w^2 G_t^2} = G_rG. \tag{38}$$

Figure 9a shows the comparison of $A_{\gamma,1}$ found numerically by running the OCP (with the settings of OCP B) for different combination of $M$ ($M \in [0.025\ 0.15]$), $G$ ($G \in [10\ 30]$) and $F_r$ ($F_r \in [0.1\ 0.2]$) and the analytical approximation given in Eq. 38. Figure 9b shows the first Fourier coefficient of elevation $\beta$ and azimuth $\phi$ as a function of the non-dimensional parameter $M_t$, as they represent the width and height of the trajectory. The analytical expression given in Eq. 36 is still a good approximation of the optimal trajectory shape.

## 5.3 Optimizing for the Mean Shaft Power considering Gravity

In this section, the on-board wind turbine induction is included in the power evaluation and the mean shaft power $\hat{P}_s$ is considered as objective function. To present the results, two different OCPs are introduced (Table 3). The mean shaft power for OCP A and B is almost identical, highlighting that one harmonic to model the productive drag and two for the roll are enough. The power for the analytical case is found by maximizing Eq. 28 with respect to $\gamma$. The figures in this section refer to OCP B.

Figure 10a shows the trajectory in the $\beta - \phi$ plane. The trajectory deviates from a circular shape, especially along the $\beta$ axis, and has a mean elevation angle of $\hat{\beta}_B = -5.5°$, higher than for the case without induction. Figure 10b shows the roll angle



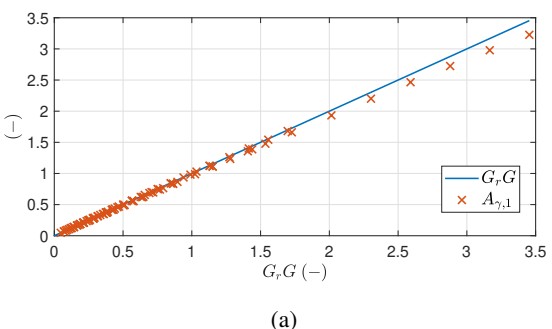
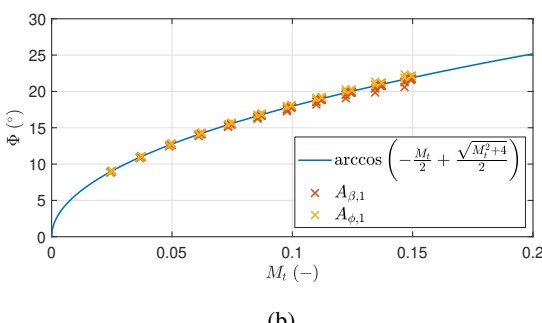

(a)                                              (b)

**Figure 9. (a)** Amplitude of the first Fourier coefficient of $\gamma$ (x) found by solving multiple OCPs and analytical approximation (-) as a function of $G_rG$ and **(b)** first Fourier coefficient of elevation $\beta$ and azimuth $\phi$ (x) found by solving multiple OCPs and analytical expression (-) as a function of the modified non-dimensional mass parameter $M_t$.

**Table 3.** Settings of the two optimal control problems maximizing the mean shaft power considering gravity.

| OCP | $N_x$ | $N_\gamma$ | $N_\psi$ | size **ov** | size **h** | $\hat{P}_s$ (kW) | $\mathcal{T}$ (s) |
|-----|-------|-----------|----------|-------------|------------|------------------|-------------------|
| A | 10 | 5 | 5 | 65 | 42 | 248.6 | 11.8 |
| B | 10 | 1 | 2 | 51 | 42 | 248.5 | 11.8 |
| L | analytical model | | | | | 272.5 | 12.6 |

as a function of the angular position in the loop. Even in this case with induction, the fluctuations are relatively small. When the AWES increases the turning radius (approximately between $-90° < \alpha < 0°$ and $90° < \alpha < -180°$ (see Fig. 10a), the roll decreases.

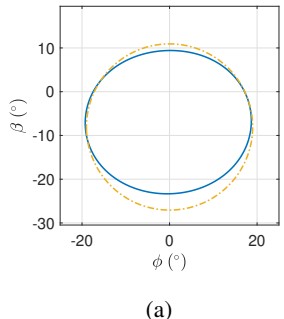
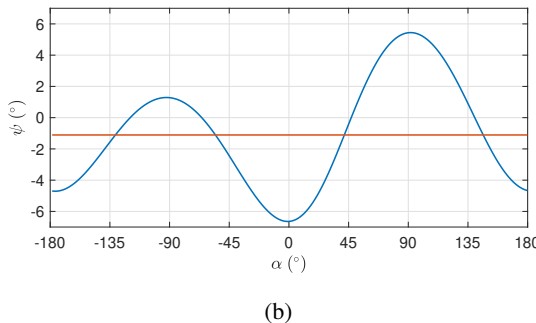

(a)                                              (b)

**Figure 10. (a)** Optimal trajectory $(-)$ and a circle with radius $\tilde{\Phi}$ (-.) and **(b)** optimal $\psi$ (blue line) and its mean (red line) as a function of the angular position.

Figure 11a shows $\gamma$ as a function of the angular position. The mean value is smaller compared to the value maximizing Eq. 28. By comparing the trends with Figure 7a, it is clear that the fluctuations of $\gamma$ are lower than in the case without induction.



The on-board wind turbine induction has a similar trend to $\gamma$, as they are linked through the expression in Eq. 28. When $\gamma$ takes negative values -in the ascendant leg- the on-board wind turbines are operated as propellers and the induction is negative. In the descendent leg, $\gamma$ takes values larger than the mean and so does the induction. Higher values of induction result in a lower

ratio between shaft power, which is the power the optimizer maximizes, and thrust power, which is the power directly linked to the dynamics. Therefore, high values of $\gamma$ are not beneficial for the shaft power production.

In Figure 11b, the AWES velocity is shown, highlighting that it fluctuates over the loop. When maximizing the mean thrust power (Sect. 5.2), the optimal AWES velocity over the loop was found to be constant. Here, it is optimal to convert part of the potential energy into electrical and part into kinetic energy, letting the velocity fluctuate over the loop.

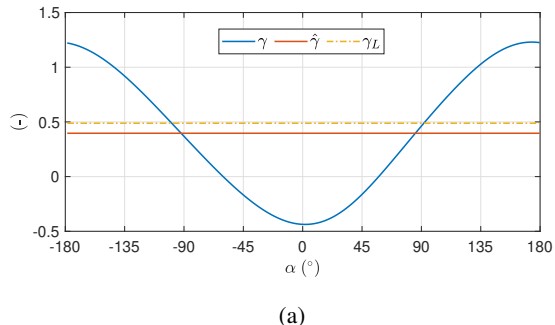

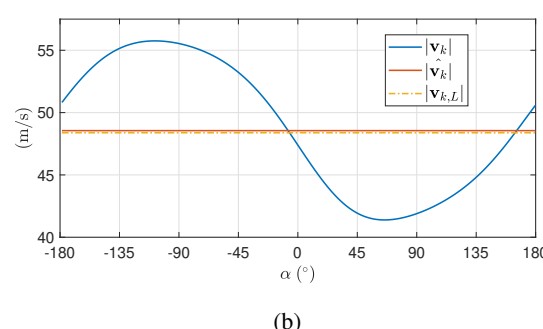

(a)                                                                                      (b)

**Figure 11. (a)** Optimal $\gamma$ and **(b)** module of the optimal AWES velocity $|\mathbf{v}_k|$ as a function of the angular position.

To conclude the analysis of the example, Figure 12a shows the shaft and the thrust power. As anticipated, when $\gamma$ takes higher values than the mean, the induction grows and the ratio between shaft and thrust power decreases consequently.

In Figure 3 the dependence of the analytical expression of the shaft power on $C_D \frac{A}{A_t}$ is shown. In Figure 12b, the dependence of the optimal mean shaft power is analyzed as a function of the same non-dimensional coefficient for three different Froude numbers (i.e. three wind speeds). In the current example, $F_r = 0.1$ corresponds to $v_w = 5.4$ m s$^{-1}$, $F_r = 0.15$ corresponds

to $v_w = 8.1$ m s$^{-1}$ and $F_r = 0.2$ corresponds to $v_w = 10.8$ m s$^{-1}$. For increasing Froude number, the solution gets closer to the analytical formulation because the power fluctuations gradually lose impact on the mean power production. Indeed, the aerodynamic forces become dominant with respect to the gravitational force.

In Sect. 5.2, it was found that the amplitude of the first Fourier coefficient of $\gamma$ for $C_D \frac{A}{A_t} = 0$ (i.e., optimizing for the thrust power) can be approximated by $A_{\gamma,1} \approx G_r G$. In Figure 13a, the trends of $\frac{A_{\gamma,1}}{G_r G}$, being $\gamma$ modeled with a single harmonic, for the

three investigated Froude numbers, are shown as a function of $C_D \frac{A}{A_t}$. For $C_D \frac{A}{A_t} \to 0$, trends are close to 1. For increasing $F_r$, the curves collapse to a unique curve. In particular, at $C_D \frac{A}{A_t} = 0.23$, which is the value for the example, the ratio $\frac{A_{\gamma,1}}{G_r G} \to 0.54$ for increasing $F_r$. For increasing values of $C_D \frac{A}{A_t}$, the ratio $\frac{A_{\gamma,1}}{G_r G}$ tends to zero, highlighting the fact that a less fluctuating value of $\gamma$ over the loop is beneficial. The plot shows also the value of $\hat{\gamma}$ as a function of $C_D \frac{A}{A_t}$, highlighting that for increasing $F_r$ the trends collapse to the value maximizing Eq. 28, indicated as $\gamma_L$.

Finally, Figure 13b shows the ratio of the first Fourier coefficient of the elevation angle $\beta$ and the azimuth angle $\phi$ with the opening angle $\tilde{\Phi}$ evaluated with Eq. 36. For $C_D \frac{A}{A_t} \to 0$, values are close to 1, as noted in Figure 9b. As $C_D \frac{A}{A_t}$ increases, the





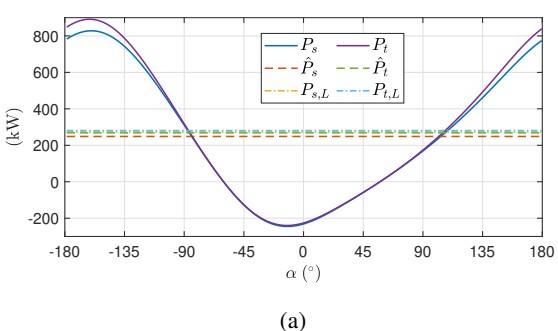

(a)

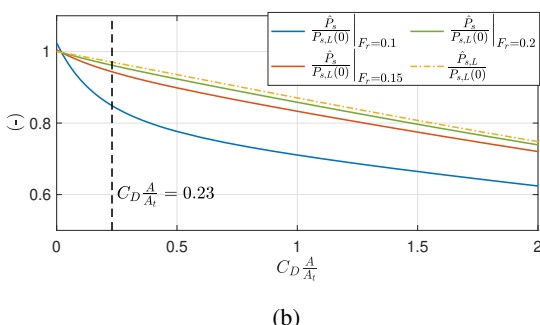

(b)

**Figure 12. (a)** Optimal shaft power production and consumption $P_s$ and thrust power $P_t$ as a function of the angular position and **(b)** optimal shaft power production normalized with the analytical expression of thrust power as a function of $C_D \frac{A}{A_t}$.

values of $A_{\beta,1}$ decrease more than $A_{\phi,1}$, showing that optimal trajectory does not have any more a circular shape and the height decreases more than the width. This effect is visible also in the example, in Figure 10a. At low Froude numbers (i.e. low wind speeds) this effect is more evident.

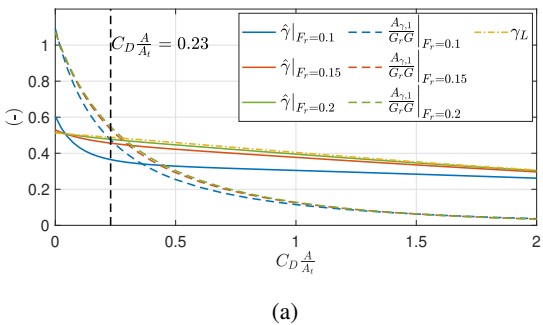

(a)

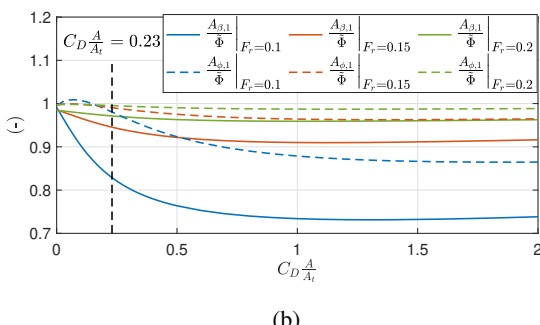

(b)

**Figure 13. (a)** Optimal values of $\hat{\gamma}$ and $A_{\gamma,1}$, normalized with $G_r G$, and **(b)** optimal values of $A_{\beta,1}$ and $A_{\phi,1}$, normalized with the analytical expression of the opening angle $\tilde{\Phi}$, as a function of $C_D \frac{A}{A_t}$ for different Froude numbers.

### 370   **5.4   Optimizing for the Mean Electrical Power considering Gravity**

In this section, the electrical efficiency is included into the optimal control problem and the mean electrical power is considered as objective function. Two OCPs, whose characteristics are given in Table 4, are introduced to present results. The power for OCP A and B is almost identical, highlighting that one harmonic to model the productive drag and two for the roll are enough. The power for the analytical case is found by maximizing Eq. 29 with respect to $\gamma$.

Figure 14a shows the trajectory in the $\beta - \phi$ plane. The trajectories deviate from the circular trajectory with opening angle $\tilde{\Phi}$ (Eq. 36), especially along the $\beta$ axis, and have a mean elevation angle of $\hat{\beta}_A = -5.1°$ and $\hat{\beta}_B = -4.9°$. Figure 14b shows





**Table 4.** Settings of the two optimal control problems maximizing the mean electrical power considering gravity.

| OCP | $N_x$ | $N_\gamma$ | $N_\psi$ | size **ov** | size **h** | $\hat{P}$ (kW) | $\mathcal{T}$ (s) |
|-----|-------|-----------|----------|-------------|------------|----------------|-------------------|
| A | 10 | 5 | 5 | 65 | 42 | 196.5 | 11.5 |
| B | 10 | 1 | 2 | 51 | 42 | 194.2 | 11.4 |
| L | analytical model | | | | | 218.0 | 12.6 |

the roll angle as a function of the angular position in the loop. As in the case maximizing mean shaft power, the fluctuations
are relatively small.

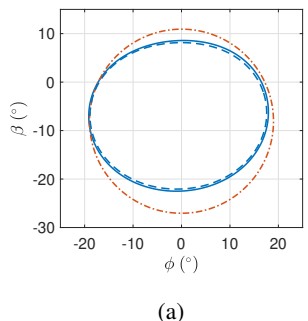

(a)

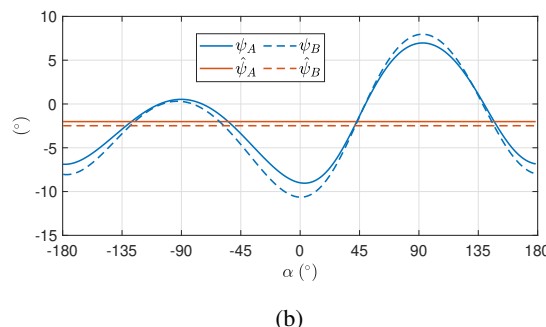

(b)

**Figure 14. (a)** Optimal trajectory for OCP A (-), B (- -) and a circle with radius $\tilde{\Phi}$ (-.) and **(b)** optimal $\psi$ as a function of the angular position.

Figure 15a shows $\gamma$ as a function of the angular position. In OCP A, the time evolution of $\gamma$ is modelled with 5 harmonics.
In the ascendant leg, $\gamma$ takes null values meaning that the power is neither spent neither consumed. Indeed, spending power
drastically reduce the overall power production because of the conversion efficiency from electrical to thrust power. This is
also highlighted by Tucker (2020). In OCP B, the time evolution of $\gamma$ is modelled with just one harmonic. The trend is however
similar to OCP A, with the minimum value being slightly negative. This means that $A_{\gamma,1}$ is similar to $\hat{\gamma}$.

Figure 15b shows the module of the AWES velocity over the loop, showing that the trend is similar for the two OCPs and,
as noted in Sect. 5.3, it is optimal to convert part of the potential energy into electrical and part into kinetic energy.

The electrical power as a function of the angular position is shown in Figure 16a. As expected when analyzing the trend
of $\gamma$, the electrical power is null in the ascendant leg and larger than the mean in the descending part. In OCP B, the mean
power is slightly lower than in OCP A but the power fluctuations are lower, which could be beneficial from a grid and power
smoothing perspective. Due to the electrical efficiency, the wind turbines are not used as propellers anymore. The results is an
even more squashed trajectory (Figure 14a) with respect to the previous Sect. 5.3 (Figure 10a) to partially limit the potential
energy exchange into kinetic and so the kite speed fluctuation.

One could try to investigate how the optimal values evolve for an increasing wind speed. Figure 16b shows the mean value
of $\gamma$ and its first Fourier coefficient $A_{\gamma,1}$ as a function of $G_r G$ (see Eq. 35 for definition) for OCP B. As noted when analyzing



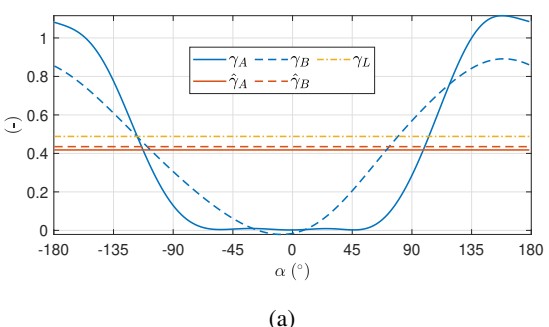
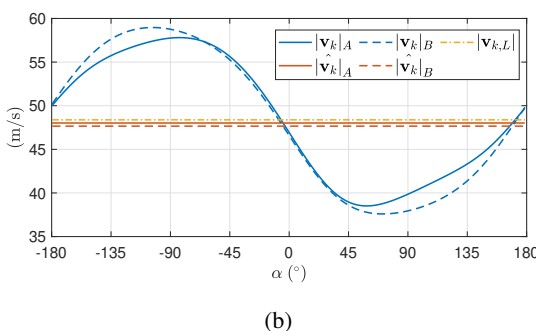

(a)                                             (b)

**Figure 15. (a)** Optimal $\gamma$ and **(b)** module of the optimal AWES velocity $|\mathbf{v}_k|$ as a function of the angular position.

Figure 15a, at $v_w = 6$ m s$^{-1}$ $A_{\gamma,1}$ is slightly larger than $\hat{\gamma}$. As wind speed increases up to approximately 8.5 m s$^{-1}$, $A_{\gamma,1}$

keeps being similar to $\hat{\gamma}$, meaning that the minimum value of $\gamma$ is close to zero and so is power ($P_{min} \approx 0$). If the wind speed increases again, $A_{\gamma,1}$ gets lower than $\hat{\gamma}$, meaning that power is always generated over the loop ($P_{min} > 0$). The main effect of the electrical efficiency on the OCP is to prevent the on-board wind turbines to be operated as propellers. Therefore, when the value of $A_{\gamma,1}$ which maximizes the shaft power $P_s$ is larger than $\hat{\gamma}$, results are expected to be modified with respect to Sect. 5.3. Instead, when $A_{\gamma,1}$ is lower than $\hat{\gamma}$ trends are expected to be equal to the analyses in Sect. 5.3. Indeed, when analyzing

Figure 16b, it is found that $\frac{A_{\gamma,1}}{G_r G} \to 0.54$ for high $F_r$. This means that for low $G_r G$, the first Fourier coefficient of $\gamma$ can be approximated with $A_{\gamma,1} \approx 0.54 G_r G$, as shown in Figure 16b.

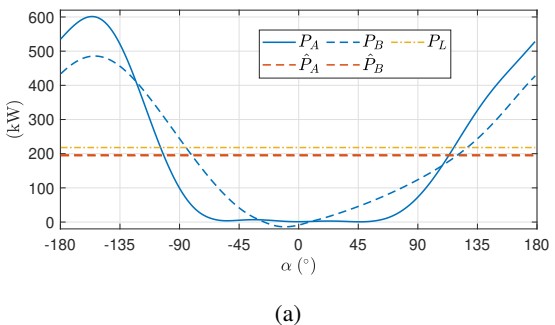
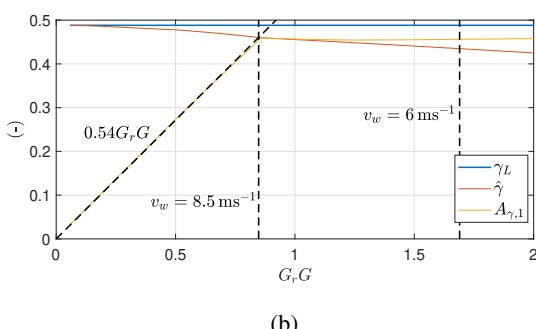

(a)                                             (b)

**Figure 16. (a)** Optimal electrical power production as a function of the angular position and **(b)** optimal values of $\hat{\gamma}$ and $A_{\gamma,1}$ as a function of $G_r G$.

In Figure 17a, the mean power normalized with the power evaluated with Eq. 29 is shown as a function of $G_r G$ for a case with $\eta_{el} = 1$, which is equivalent to the case in Sect. 5.3, and for a case with $\eta_{el} = 0.8$, as in this section. The two curves for wind speed lower than 8.5 m s$^{-1}$ diverges. A low electrical efficiency $\eta_{el}$ not only decreases the power output as in Eq. 29, but

also decreases the efficiency with respect to the analytical approximation due to its effect on the dynamics.

To conclude, Figure 17b shows the evolution of the first Fourier coefficient of the elevation angle $A_{\beta,1}$ and of the azimuth $A_{\phi,1}$ as a function of $G_r G$. For high wind speed (i.e. low $G_r G$), their value is similar to the approximation given in Eq. 36. As




$G_rG$ increases, for $\eta_{el} = 1$, $A_{\phi,1}$ stays almost constant while $A_{\beta,1}$ decreases. Smaller $A_{\beta,1}$ means smaller vertical height, which results in lower potential energy converted into electrical and kinetic energy over the loop. For $\eta_{el} = 0.8$ after $v_w = 8.5 \text{ m s}^{-1}$,

both Fourier coefficients decrease rapidly meaning that smaller loops are performed. Smaller loops are therefore beneficial at low wind speed as they decrease the energy fluctuations.

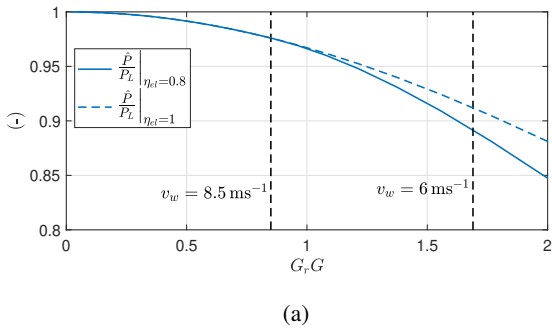

(a)

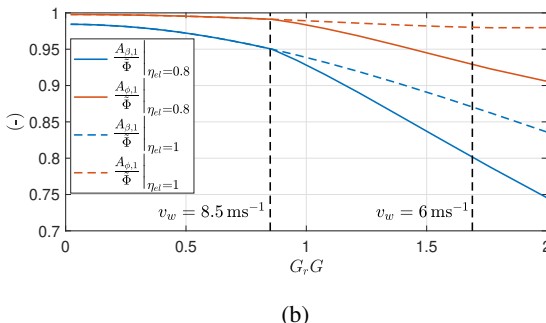

(b)

**Figure 17. (a)** Normalized electrical power for a case with $\eta_{el} = 0.8$ (-) and $\eta_{el} = 1$ (- -) and **(b)** optimal values of $A_{\beta,1}$ and $A_{\phi,1}$ normalized with the analytical expression of the opening angle $\tilde{\Phi}$ for a case with $\eta_{el} = 0.8$ (-) and $\eta_{el} = 1$ (- -) as a function of $G_rG$.

## 6     Optimal Control Problem considering Gravity, Wind Shear and Elevation Constraint

In this section, the wind shear is included in the problem. The reference altitude is taken $h_0 = 100$ m, the wind shear exponent $\alpha_s = 0.2$ and the reference wind speed $v_{w,0} = 6 \text{ m s}^{-1}$. To make the problem more realistic, a constraint on the minimum

elevation angle of $\beta_m = 10°$ (which is equivalent to a constraint on the minimum flight altitude) is included. The value of $\beta_s$ needed to compute $\mathbf{e}_3$ and then the spanwise unit vector $\mathbf{s}$ (Eq. 5) is taken as the mean elevation angle $\beta_s = \hat{\beta}$. With this definition, the case of no roll ($\psi = 0$) is obtained when the wing span is in the plane perpendicular to the mean elevation angle, as in Trevisi et al. (2021).

Two OCPs are solved and they are summarized in Table 5. OCP A features 5 harmonics to model the control inputs, while

OCP B has one harmonic to model the on-board wind turbine thrust and two for the roll, as in the previous sections. The two optimizations have similar mean electric power outputs, meaning that they are almost equivalent.

**Table 5.** Settings of the two optimal control problems maximizing the mean power considering gravity and wind shear.

| Case | $N_x$ | $N_\gamma$ | $N_\psi$ | size **ov** | size **h** | $\hat{P}$ (kW) | $\mathcal{T}$ (s) |
|------|-------|------------|----------|-------------|------------|----------------|-------------------|
| A | 10 | 5 | 5 | 65 | 42 | 115.3 | 11.5 |
| B | 10 | 1 | 2 | 51 | 42 | 113.3 | 11.4 |





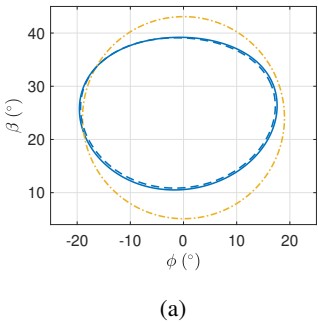

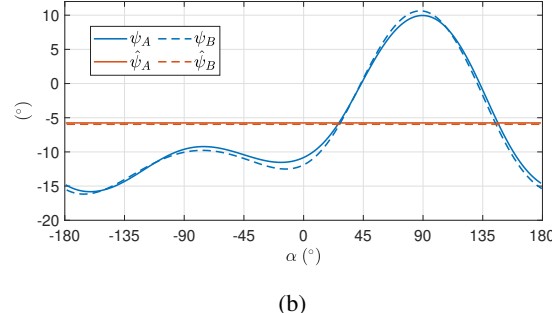

**Figure 18. (a)** Optimal trajectory for case A (-), B (- -) and a circle with radius $\Phi$ (-.) and mean elevation angle $\arctan\sqrt{\alpha_s}$ and **(b)** optimal $\psi$ as a function of the angular position.

Figure 18a shows the trajectory for OCP A and B and compare it with a circle of radius $\tilde{\Phi}$ centered at an elevation of $\hat{\beta} = \arctan\sqrt{\alpha_s}$, as this formulation identifies the elevation of the center of the wind power window (Argatov et al. (2011)). As for the cases analyzed in Sect. 5.3 and 5.4, the trajectory is squashed along the vertical direction. The constraint on the
minimum elevation angle is not used, as the trajectory of both cases is always strictly higher than $\beta_m = 10°$. The roll angle $\psi$ is shown in Figure 18b. In *LT-GliDe* (Trevisi et al. (2021)) the flight stability of AWES is studied by linearizing the equations of motion with respect to a fictitious steady state condition, where the AWES moves in a circular trajectory with a constant velocity. This steady state is characterized by the kite span being perpendicular to the mean elevation angle direction. In this section, this condition is identified by $\psi = 0$. The roll fluctuations, shown in Figure 18b, are limited in amplitude and might be
considered within the linear bounds of the linearization validity of *LT-GliDe*. More analyses to prove this will be carried out in future works.

Figure 19a shows the evolution of the on-board wind turbines thrust as a function of the angular position for the two cases. The trends are similar to the analyses in Sect. 5.4. It is optimal to use the on-board wind turbines only to generate power and not as propellers. Even if the trends of $\gamma$ for OCP A and B are quite different, the overall power production is similar,
meaning that power production is not sensitive to harmonics of $\gamma$ higher than one. Figure 19b shows the wind speed that the AWES encounters over the loop due to the wind shear. Clearly, at the top of the loop ($\alpha = 90°$), the wind speed is the highest, sweeping approximately 1.5 m s$^{-1}$ over the trajectory. In this section, the mean wind speed over the loop is used to evaluate the Froude number $F_r$ (Eq. 34) and consequently the gravity ratio $G_r$ (Eq. 35). These numbers will be used later in this section to generalize results.

Figure 20a shows the AWES velocity as a function of the angular position. As discussed in the previous sections, in the descending leg the AWES convert the potential energy into electrical, producing power, and kinetic energy, accelerating. In the climbing leg instead electrical power is not spent and kinetic energy is transformed into potential. Figure20b shows the power production as a function of the angular position. When looking at power and tether force (not shown here as it follows the AWES velocity trend squared) to characterize the operations of a real system, the maximum power and the maximum and





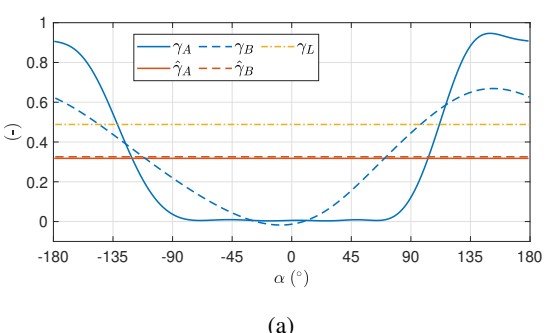 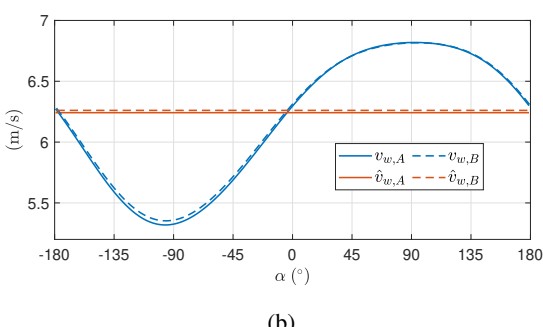

(a) (b)

**Figure 19. (a)** Optimal $\gamma$ and **(b)** wind velocity as a function of the angular position.

minimum tether force would be constrained not to exceed some given values. To properly include these constraints, additional control inputs, useful to model the de-powering of the AWES (e.g. the lift coefficient), shall be considered in the analysis.

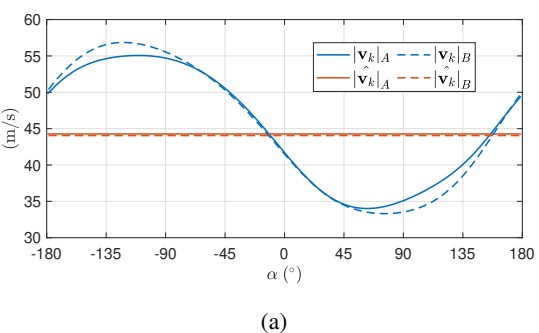 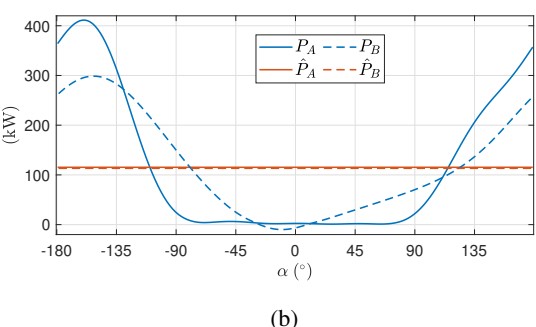

(a) (b)

**Figure 20. (a)** Module of the optimal AWES velocity $|\mathbf{v}_k|$ and **(b)** optimal power production $P$ as function of the angular position.

As carried out in the previous section, trends are studied as a function of the Froude number for the optimal control problem B. Figure 21a shows the dependence of $\hat{\gamma}$ and $A_{\gamma,1}$ as a function of $G_r G$. $\hat{\gamma}$ decreases when $G_r G$ increases (i.e., the wind speed decreases). At low wind speed, $\hat{\gamma}$ takes low values so that the AWES speed over the loop is higher, which is beneficial to stay

airborne. $A_{\gamma,1}$ for low $G_r G$ has a linear trend, as noted in Sect. 5.4. When $A_{\gamma,1}$ is equal to $\hat{\gamma}$, the minimum power production over the loop is null. For lower wind speeds (i.e., higher $G_r G$), it is not optimal anymore to increase $A_{\gamma,1}$ because the on-board wind turbines would be used as propellers with a high penalty on the mean power production. To conclude, Figure 21b shows the ratio of the first Fourier coefficient of $\beta$ and $\phi$ with respect to the analytical expression of the opening angle $\tilde{\Phi}$ for a case maximizing electrical power ($\eta_{el} = 0.8$) and shaft power ($\eta_{el} = 1$). After the cusp in Figure 21a, the two trends diverge and for

$\eta_{el} = 0.8$ smaller loops are optimal so that the exchange of potential energy over the loop is reduced.





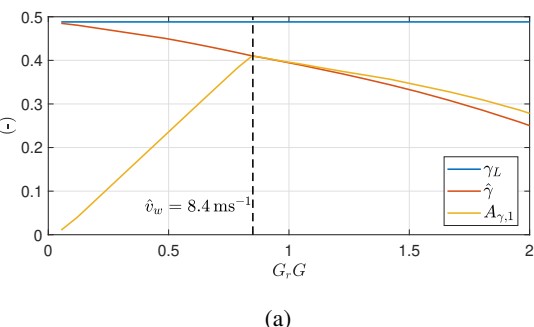

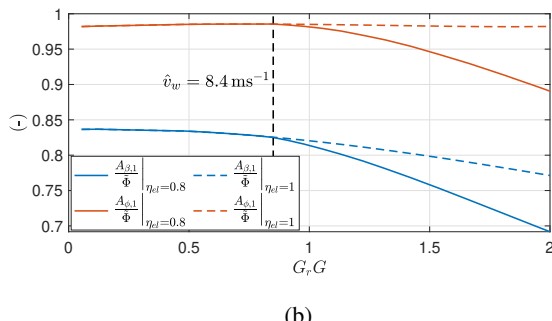

(a)     (b)

**Figure 21. (a)** Optimal values of $\hat{\gamma}$ and $A_{\gamma,1}$ and **(b)** optimal values of $A_{\beta,1}$ and $A_{\phi,1}$ normalized with the analytical expression of the opening angle $\tilde{\Phi}$ for a case with $\eta_{el} = 0.8$ (-) and $\eta_{el} = 1$ (- -) as a function of $G_r G$.

## 7 Conclusions and Discussion

In this work, a novel methodology to study optimal trajectories for Fly-Gen AWES is introduced. The chosen low-fidelity dynamic model is characterized by two degrees of freedom (the AWES is modelled as a point mass with constant tether length) and two control inputs. The degrees of freedom are the elevation and the azimuth angle. The control inputs are the
roll angle, defined as the rotation around the relative velocity direction, and the on-board wind turbines thrust coefficient. An Optimal Control Problem is formulated in the frequency domain through a Harmonic Balance method. Working with the Fourier coefficients of the time series, instead of the time series themselves, allows to reduce the problem size, to implicitly impose periodicity and to gain an intuitive understanding of the results by analyzing the harmonic contributions. Moreover, the analytical gradient of the objective function and the constraints with respect to the optimization variables can be provided to
the solver, allowing for a deep and fast convergence of the optimal solutions.

The *MX2* design from Tucker (2020) is taken as a reference AWES to introduce the results. To isolate the effects of each physical phenomenon, results are presented with an increasing level of complexity from the most idealized case and they are compared with analytical solutions from literature, whenever possible. A set of idealized case studies with no constraint on the minimum elevation angle and uniform wind inflow are initially studied. If gravity is neglected, the solution is steady and it
can be described by analytical expressions. If gravity is considered, three different optimal control problems, characterized by three different objective functions, are solved:

i) If the mean thrust power (mechanical power neglecting on-board wind turbines induction) is the objective function, the optimal trajectories are circular, have a constant AWES velocity and the wing span is perpendicular to the incoming wind. To obtain this condition, all the potential energy is converted into electrical by the on-board wind turbines. At low wind speed,
on-board wind turbines are then used as propellers in the ascendant part of the loop. The optimal power, the trajectory shape and the production strategy can be accurately approximated with analytical expressions;

ii) If the mean shaft power (mechanical power considering on-board wind turbines induction) is the objective function, the potential energy, in the descending leg, is partially converted into electrical and partially into kinetic energy. This is because





the power conversion penalizes solutions with high on-board wind turbines induction. Therefore, the velocity fluctuates over
the loop and the trajectories are squashed along the vertical direction to decrease the potential energy exchange;

iii) If the mean power electrical provided to the grid is the objective function (i.e. the electrical efficiency is included), the
on-board wind turbines never operate as propellers. If operated as propellers, power would be converted from mechanical
into electrical while descending and from electrical into mechanical while ascending, leading to large power losses due to the
electrical efficiency. This effect is found only at low wind speed, when propelling the AWES in the climbing leg maximizes the
mean shaft power. Past a given wind speed, using the on-board wind turbines as propellers does not maximize the mean shaft
power and the influence of the electrical efficiency on the production strategy vanishes.

When the wind shear and a constraint on the minimum elevation angle are included in the optimal control problem for
maximizing the electrical power, trends are similar to what found in the case with uniform inflow. Therefore, the power
production strategy does not heavily depend on the wind shear. For the analyzed example, the constraint on the minimum
elevation angle is not active.

For all the analyzed cases, additional analytical approximations characterizing the solution are introduced. These approxi-
mations are found by modelling the control inputs with the lowest number of harmonics. The on-board wind turbines thrust
can be modelled with just one harmonic and the roll with two harmonics without loss of generality of the results.

The results of this work align with the discussions in Tucker (2020). Moreover, the results presented in this work have a
strong mathematical foundation, as the trajectory and the control inputs are found by solving optimal control problems. These
methods are planned to be applied, with appropriate modifications, to other AWE architectures and to other trajectory types.
A comparison between circular and figure of eight trajectories is foreseen. Finally, the physical understanding and methods
proposed here are envisaged to be incorporated into the design, analysis and optimization framework *T-Glide* (Trevisi et al.
(2022)), with the aim of improving the power estimation and including an optimal control module.

**Appendix A: Figures of Comparison between Frequency-Domain Formulation and Time Integration**

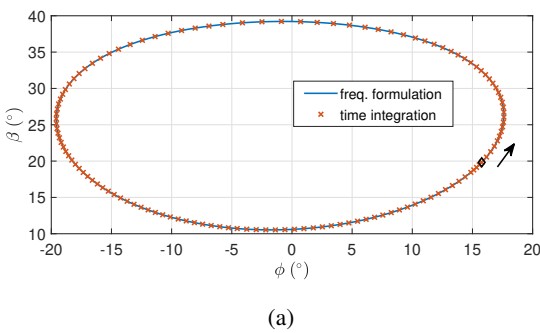 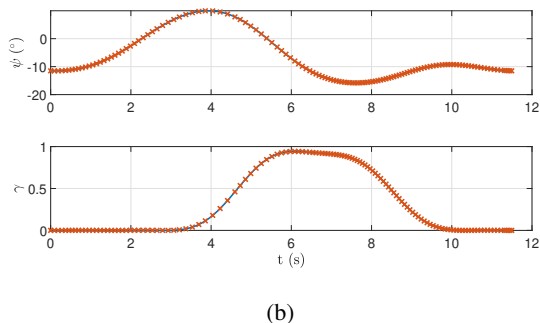

(a)                                                                 (b)

**Figure A1. (a)** Azimuth and elevation of the trajectory found with the harmonic balance method and the time integration scheme for a circular shaped trajectory and **(b)** time series of the control inputs provided to the harmonic balance method and the time integration scheme for a circular shaped trajectory.

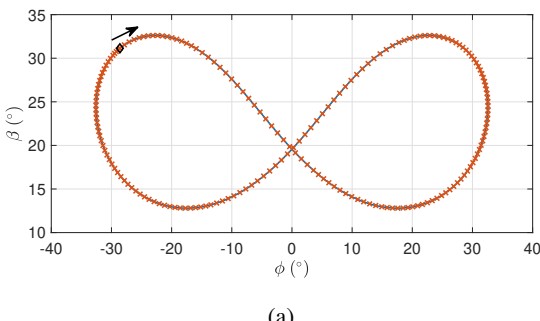 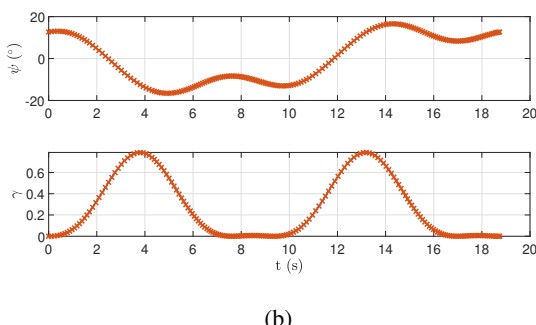

(a)                                                                 (b)

**Figure A2. (a)** Azimuth and elevation of the trajectory found with the harmonic balance method and the time integration scheme for a figure of eight shaped trajectory and **(b)** time series of the control inputs provided to the harmonic balance method and the time integration scheme for a figure of eight shaped trajectory.

*Author contributions.* FT, ICF, and GP conceptualized the study and the research methods. FT developed the research methods. FT and ICF developed the code. FT produced the results and wrote the draft version of the paper. CEDR and AC supervised. ICF, GP, CEDR and AC reviewed the draft version.

*Competing interests.* The authors declare that they have no conflict of interest.

*Acknowledgements.* The work by ICF was carried out under the framework of the GreenKite-2 project (PID2019-110146RB-I00) funded by MCIN/AEI/10.13039/501100011033.



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
