# Peer review of "Flight Trajectory Optimization of Fly-Gen AWE Systems through a Harmonic Balance Method"

_Wind Energy Science, 2022_

## Author Comment (AC1)

Dear Editor, dear Reviewers,

thank you for your comments and for taking the time to review our work.

In the following we go through your *comments* and provide, for each one, both our *responses* and the *actions* we have taken to accommodate your feedback in the revised manuscript.

Best regards,

The Authors

**Reply to Anonymous Referee #1**

[https://doi.org/10.5194/wes-2022-49-RC1]

**Review Comment**

The paper considers the optimization of flight trajectories of Fly-Gen AWES using the Harmonic Balance method. The methodology is presented clearly and the comparison of the various optimizations undertaken by the authors as well as the subsequent discussions provides a novel contribution. The detailed analysis in sections five and six provides a good foundation for future research and adoption of the Harmonic Balance method in AWES trajectory optimization.

I recommend minor copy editing and, for completeness, all variables (e.g. Gt and gamma) should be introduced prior to being used. I have found no technical errors, however, Figure 3a is difficult to read and should be redone.

**Authors Answer**

The authors would like to thank the reviewer for the time dedicated to our paper.

We went through the whole text again and introduced those variables that were not well defined. Moreover, Fig 3a has been redone to be more readable now.

**Authors Actions**

We reviewed the text to include better some variables and updated Fig 3a accordingly. ted in the final paper.

**Reply to Anonymous Referee #2**

[https://doi.org/10.5194/wes-2022-49-RC2]

**Review Comment**

The paper presents a novel methodology to study the trajectories of Airborne Wind Energy Systems (AWES) solving Optimal Control Problems (OCP) in the frequency domain. Optimizing the Fourier coefficients instead of the time series allows to reduce the size of the problem and imposes periodicity.

The authors then proceed to solve several OCP with different objective functions and with an increasing level of complexity. Starting with a theoretical ideal case (perfectly crosswind flight with constant wind speed and ignoring the effects of gravity and elevation angle limitations) and concluding with a case study that takes into consideration wind shear, elevation angle limitations, the effects of gravity and electrical power losses.

The methodology is clearly explained, is a good contribution for the development of research in AWES and it promises good results in the future for other concepts of AWES and other trajectory shapes.

I would only add a table summarizing the nomenclature used throughout the paper.

**Authors Answer**

The authors would like to thank the reviewer for the time dedicated to our paper.

We fully agree that a nomenclature table is missing in this paper. Thank you again.

**Authors Actions**

We added the nomenclature section at the end of the paper.

**Reply to Anonymous Referee #3**

[https://doi.org/10.5194/wes-2022-49-RC3]

**Review Comment**

This paper contains two contributions. First, it is the first academic work where the harmonic balance method is applied to compute power-optimal trajectories of an AWE system. As a case study, the authors apply the method to a low-fidelity point-mass model from the literature, adapted here to model Fly-Gen AWE systems. Second, the method is applied to compute power-optimal trajectories for varying objectives and constraints, in order to investigate the influence of different physical effects (e.g. electrical efficiency of on-board turbines, gravity,...) on the trajectories, and compare them to idealized analytic solutions from the literature. The paper is well-structured. The authors provide a good overview of the available literature. The harmonic balance method is an interesting candidate method for AWE trajectory optimization. The authors present an efficient problem formulation while always checking for accuracy (discretization sensitivity is computed for all examples). It provides a good basis for further work on different system types, trajectory shapes and maybe also higher fidelity models. The thorough and illuminating analysis of different Fly-Gen trajectories is a fine illustration of the power of optimal control in AWE system dynamic analysis.

The paper is technically sound. Here are some suggestions to improve the quality of the manuscript:

**Authors Answer**

The authors would like to thank the reviewer for the time dedicated to our paper. Suggestions were greatly appreciated and helped us to improve this paper. Below are our various comments.

**Authors Actions**

None

Review Comment 1 Introduction
* * *
L. 82-83: It is stated (also e.g. in the abstract) that working with Fourier coefficients instead of a time series allows to "reduce the problem size significantly". Although the intuition behind this statement is clear (particularly for Fly-Gen), there is no previous work on this, nor do the authors provide a comparison with a time-domain OCP of similar accuracy. In the example, an HB-OCP is solved with  $N_x = 10$ , leading to  $(2*N_x + 1) nx = 21$  times nx variables in the NLP. For a time period of ~12 s, one can imagine a time-domain multiple-shooting NLP with 20 intervals (and identical control parametrization) that solves this problem with similar accuracy. The relative performance of harmonic balance will be highly problem-dependent. Hence my suggestion is to weaken the claim: "... to potentially reduce the problem size significantly depending on the problem at hand". Or, of course, to include a comparison, which would be highly interesting.

In my opinion, the main benefit of the harmonic balance method lies in how it facilitates interpretation of the optimal result, as illustrated in this paper.

In any case, an indication of computation times would be an interesting addition to the text.

**Authors Answer**

We agree with this point and we modified the manuscript accordingly in some points.

We also feel that a comparison with time-domain OCP would be highly interesting, but we prefer to leave this study, which might take a few pages and a consistent amount of work, for future works. Through the paper, we performed a sensitivity on the order of the control inputs series, which led us to the derivation of analytical approximations. However, we did not perform a detailed sensitivity of the order of the state variables series because the computational time and our research questions were not requiring this study. We took a number high enough ( $N_x = 10$ ) to be sure to accurately model the dynamics. However,  $N_x$  could be reduced without modifying the solution. For OCP A in Section 6,  $N_x$  could be taken equal to 6 (it needs to be larger than the order of the control inputs series) and, for OCP B,  $N_x$  could be taken equal to 4. For this last case, the number of parameters modelling the state space is nx=9, which in the equivalent time-domain problem would correspond to intervals of approximately 45 degrees for circular trajectories. We look very much forward to a detailed comparison.

**Authors Actions**

We added "potentially" in the abstract, at line 4. We replaced "reduce the problem size significantly" with "potentially reduce the problem size significantly depending on the problem at hand" at line 83, as suggested. We added the information about computational time at line 211. We added "potentially" in the conclusions, at line 465.

**Review Comment**

2.2. Frequency Domain Formulation

\_\_\_\_\_

- see first comment

L. 159: "the capability of solving for both stable and unstable (unlike time integration methods) branches of periodic solutions in an efficient way". It is unclear what is meant here. In the time domain, also unstable periodic orbits can be computed efficiently with multiple shooting or direct collocation transcriptions.

**Authors Answer:**

We agree with this point.

**Authors Actions**

We removed the text inside the brackets: "(unlike time integration methods)" at line 160.

**Review Comment**

L181-189: This is a crucial paragraph (derivation of Eq. 23). All the steps that are now described only in words should be written here in formulas to allow for re-implementation by others.

**Authors Answer**

Thanks for pointing this out. All the necessary equations were actually written but with a wrong notation.

**Authors Actions**

We modified eq 23 accordingly.

**Review Comment**

L. 204-206: Given that with algorithmic differentiation (and the ubiquitous AD tools), you can get the gradient of any function for free, I would not advertise analytic gradients as a significant advantage of the harmonic balance method.

Authors Answer We also agree with this point.

Authors Actions We removed "One of the advantages of the frequency formulation is that" at line 204.

**Review Comment**

5.2 Optimizing for the Mean Thrust power considering Gravity
* * *
L. 284-285: The absolute value of the AWES velocity is imposed to be constant. It is unclear why? Also, in L. 348 it says that in Section 5.2, the AWES velocity was found to be constant? Please clarify if it is imposed (and why) or if it is found to be optimal to be constant.

**Authors Answer**

In OCP B, we impose the norm of AWES velocity over the loop to be constant by constraining its first Fourier coefficients to be 0. In OCP A, the norm of the AWES velocity v varies by approximately 4 m/s over the loop, so from this case we cannot state that optimal trajectories are characterized by a constant v.

Showing that OCP A and OCP B are equivalent, we find that the optimal solution is characterized by a constant v over the loop.

In other words, with this study we find that the sensitivity of the objective function with respect to the Fourier coefficients of v is really small. We thought that this way of presenting the results was the most educational.

**Authors Actions**

We modified the notation (line 258 and Table 2) and terminology (absolute value-> norm) and added a sentence at line 306.

Review Comment: L. 313: OPC --> OCP

Authors Answer: Thanks

**Authors Actions**

We modified it accordingly.